# Root Characteristics and Water Erosion-Reducing Ability of Alpine Silver Grass and Yushan Cane for Alpine Grassland Soil Conservation

**Jung-Tai Lee \*, Shun-Ming Tsai, Yu-Jie Wu, Yu-Syuan Lin, Ming-Yang Chu and Ming-Jen Lee**

Department of Forestry and Natural Resources, National Chiayi University, Chiayi 60004, Taiwan; ssmart0110tw@gmail.com (S.-M.T.); s1052152@mail.ncyu.edu.tw (Y.-J.W.); s1070176@mail.ncyu.edu.tw (Y.-S.L.); s1060103@mail.ncyu.edu.tw (M.-Y.C.); mjlee@mail.ncyu.edu.tw (M.-J.L.)
**\*** Correspondence: jtlee@mail.ncyu.edu.tw; Tel.: +886-5-271-7482

**Abstract:** In Taiwan, intensive forest fires frequently cause serious forest degradation, soil erosion and impacts on alpine vegetation. Post-fire succession often induces the substitution of forest by alpine grassland. Alpine silver grass (*Miscanthus transmorrisonensis* Hay.) and Yushan cane (*Yushania niitakayamensis* (Hay.) Keng f.) are two main endemic species emerging on post-fire alpine grassland. These species play a major role in the recovery of alpine vegetation and soil conservation of alpine grassland. However, their root traits, root mechanical properties and water erosion-reducing ability have still not been well studied. In the present study, root characteristics were examined using a complete excavation method. Root mechanical characteristics were estimated by utilizing the uprooting test and root tensile test, and hydraulic flume experiments were performed to investigate the water erosion-reducing ability using 8-month-old plants. The results show that the root architecture system of Alpine silver grass belongs to fibrous root system, while the Yushan cane has sympodial-tufted rhizomes with a fibrous root system. Root characteristics reveal that relative to Alpine silver grass, Yushan cane has remarkably larger root collar diameter, higher root biomass, larger root volume, higher root density, and a higher root tissue density. Furthermore, uprooting resistance of Yushan cane is notably higher than that of Alpine silver grass. However, the root tensile strength of Alpine silver grass is significantly higher than that of Yushan cane. Additionally, hydraulic flume experiments reveal that Yushan cane has significantly lower soil detachment rates than that of Alpine silver grass. Collectively, these findings clearly show that Yushan cane has superior root characteristics and water erosion-reducing ability than Alpine silver grass and is thus more suitable for the conservation of alpine grassland.

**Keywords:** alpine grassland; root biomechanics; root system; soil stability; water erosion-reducing ability

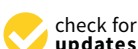



## 1. Introduction

Taiwan, a subtropical maritime island in Southeast Asia, is rich in alpine forest resources. Due to harsh drought conditions and high inaccessibility, alpine forests are very vulnerable to forest fires caused by lightning strikes [1]. Severe forest fires often cause significant forest deterioration and vegetation succession [2,3]. In addition, wildfires can damage forest soil texture, and trigger landslides and runoff erosion after intense rainstorms [4,5]. Usually, wildfires burn out forest vegetation, and induce primary ecological succession. Previous studies have indicated that post-fire forest succession occurs with the substitution of dominant herbs and shrubs in alpine vegetation [6,7]. In Taiwan, Alpine silver grass (*Miscanthus transmorrisonensis* Hay.) and Yushan cane (*Yushania niitakayamensis* (Hay.) Keng f.) are two endemic alpine pioneer species emerging on post-fire alpine grassland [8]. Generally, pioneer alpine vegetation plays an essential role in vegetation restoration and soil conservation of alpine grasslands [9,10].



The crucial functions of roots are anchorage and absorption of water and nutrients. Plant root systems can significantly affect slope stability and soil erosion [11]. Patterns of root system architecture are affected by plant species and environments, and can be categorized into seven types, i.e., P- (parallel), H- (horizontal), PH- (parallel and horizontal), R- (right), VH- (vertical or horizontal), V- (vertical) and M- (massive) type [12]. Alpine environments are characterized by dry, cold, windy, and snowy conditions. Alpine plants develop diverse root morphology and properties to survive and thrive in the harsh environmental conditions of higher altitudes [13,14]. Root system architecture plays a vital role in anchorage. Mickovski et al. [15] indicated that Vetiver grass (*Vetiveria zizanioides* L.) has an M-type fibrous root system that can resist uprooting and torrential runoff. Root traits also significantly influence its biomechanical properties. It has been demonstrated that ultimate pullout resistance is highly correlated with root collar diameter [16]. Lee et al. [17] reported that tree uprooting resistance is highly influenced by root collar diameter and root biomass. Hudek et al. [14] also demonstrated that alpine grasses have equivalent root tensile strength to small shrubs. Several studies have demonstrated that Poaceae species with M-type fibrous roots have higher soil binding capability and can significantly enhance the slope stability of alpine lands [18–20]. Previous studies also demonstrated that fibrous roots are more efficient than tap roots in preventing concentrated flow erosion [21–24]. Furthermore, root functional traits have a great influence on soil erosion. Several studies reported that the soil erosion rate is negatively correlated to root density and root length density [21,25]. Gyssels and Poesen [26] indicated that flow erosion rates decrease with increasing root density. In addition, Burylo et al. [27] showed that the soil erosion rate is closely correlated with fine roots. However, there have been very few studies on the root characteristics and water erosion-reducing ability of Alpine silver grass and Yushan cane. The hypothesis of this study is that Yushan cane with perennial symoidial rhizomes and profuse fibrous lateral roots could exhibit better ability in control of soil erosion and degradation than Alpine silver grass. Since Alpine silver grass and Yushan cane are the major dominant species on post-fire grassland, this study is aimed to investigate and compare the root traits, root mechanical properties and water erosion-reducing ability of these two native alpine pioneer species in relation to post-fire vegetation restoration and soil erosion control of alpine grasslands.

## 2. Materials and Methods

### 2.1. Site and Vegetation

The sampling site is located at the Hohuan Mountain, 3000 m a.s.l., at 24°09′34″ N, 121°17′42″ E in Taiwan, an alpine grassland (Figure 1). The annual mean temperature in the region is 7 °C, and the annual rainfall is about 3500 mm [8]. This 500 ha research area, with a downward slope in the northeast direction and average slope of 10°, is covered by 46% dominant Alpine silver grass (*Miscanthus transmorrisonensis*) and 42% Yushan cane (*Yushania niitakayamensis*) vegetations (Figure 2). The soil profile of Yusan cane grassland showed an Umbric epipedon (A, at 0–22 cm) and two Cambic horizons (Bw1, at 22–32 cm and Bw2, at 32–44 cm). The soil pedon has two main lithological discontinuities at about 40 and 70 cm below soil surface. The type of soil was classified as typic Haplumbrept, fine, illitic, frigid and inclinic. The soil materials of upper 45 cm and 45–70 cm are silty clay and sandy loam, respectively [28].

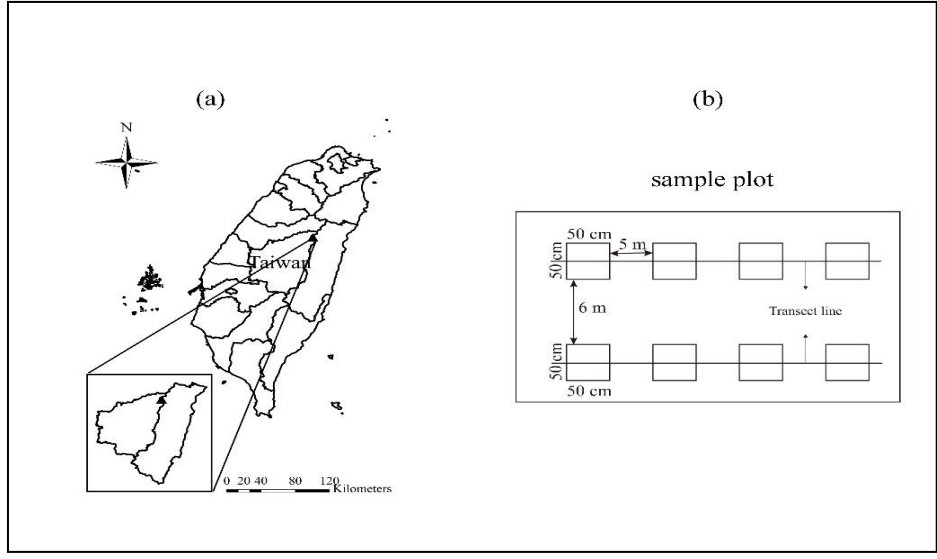

**Figure 1.** Diagram of experimental plot location (**a**) and the sample plot layout (**b**).

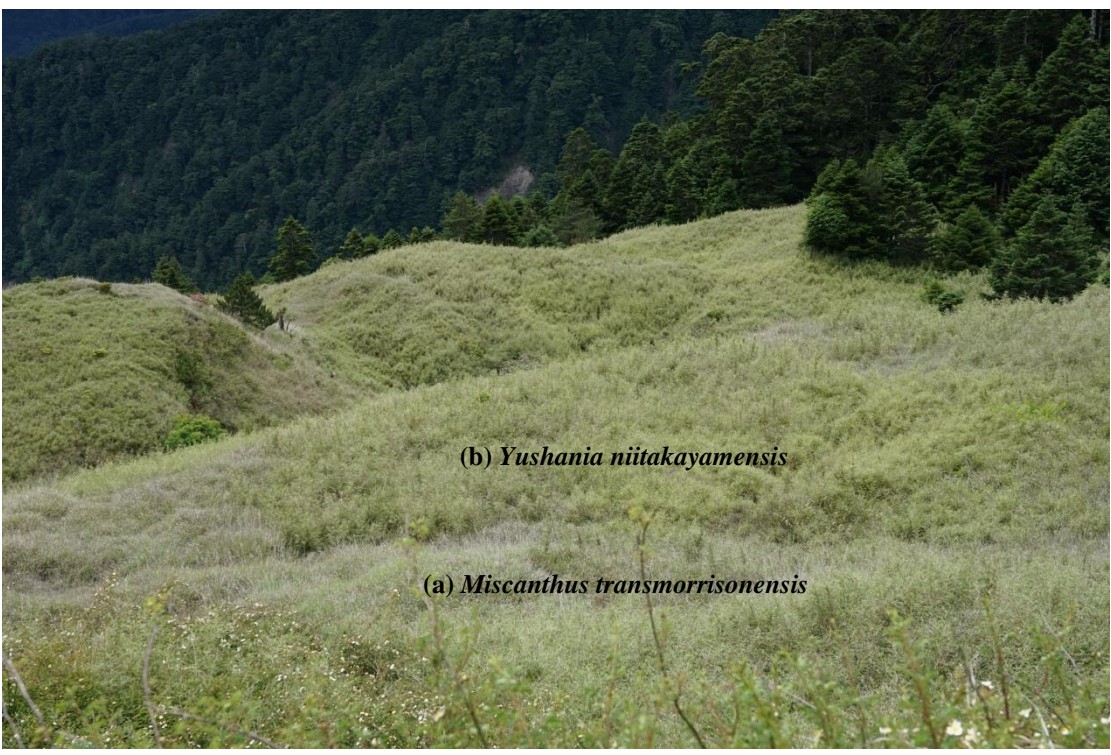

**Figure 2.** The distribution of *Miscanthus transmorrisonensis* (**a**) and *Yushania niitakayamensis* (**b**) at Hohuan Mountain alpine grassland. Photo taken by M.-Y. Chu in July 2019.

### 2.2. Plant Raising

Four 50 cm × 50 cm-quadrat sample plots at 6 m intervals along two transects lines were set up at Alpine silver grass and Yushan cane vegetations at the Hohuan Mountain alpine grassland in July 2019, respectively. One thousand ramets of each species were randomly selected, carefully excavated and collected by hand with a small steel shovel. The average height of ramets for Alpine silver grass and Yushan cane were $4.6 \pm 1.4$ cm and $5.2 \pm 1.6$ cm, respectively. Ramets of each species were wrapped in wet paper towels and put in polyethylene bags and stored in portable ice box to conserve the viability. Then, the ramets were brought back to the laboratory and planted into high crates (30 × 30 × 50 cm,

L × W × H) and short crates (30 × 30 × 30 cm, L × W × H). Before planting, the crates were packed with alpine soils collected from the alpine grassland. The chemical and physical properties of the soil are shown in Tables 1 and 2. The organic carbon and total nitrogen contents were analyzed by combustion method using Vario EL cube elemental analyzer. In addition, Mehlich 3 extragent (composing 0.2 M glacial acetic acid, 0.25 M ammonium nitrate, 0.015 M ammonium fluoride, 0.013 M nitric acid, and 0.001 M ethylene diamine tetraacetic acid (EDTA)) was used to extract the elements, such as P, K, Ca, Mg, Zn, Mn, Fe, Cu, Cd, Cr, Ni, and Pb. The concentrations of these elements were analyzed using an inductively coupled plasma optical emission spectroscopy (ICP-OES) method. For root traits and biomechanical study, 24 ramets of each species were transplanted to 24 high crates separately. For hydraulic flume experiment, 36 ramets of each species were transplanted to 24 short crates, respectively. Meanwhile, twelve bare soil crates served as control. All crates were randomly arranged in a nursery under an external environment and watered once every three days. Crates were rotated weekly to reduce shielding effect. Altogether, the azimuth of crate was unaltered.

**Table 1.** Chemical characteristics of soil used in this research.

| Properties | Soil |
|---|---|
| pH (water) | 6.21 |
| Conductivity (dS m$^{-1}$) | 0.11 |
| Organic carbon (g kg$^{-1}$) | 1.95 |
| Total nitrogen (%) | 0.27 |
| Phosphorus (mg kg$^{-1}$) | 20 |
| Potassium (mg kg$^{-1}$) | 78 |
| Calcium (mg kg$^{-1}$) | 1359 |
| Magnesium (mg kg$^{-1}$) | 162 |
| Zn (ppm) | 6.87 |
| Mn (ppm) | 143 |
| Fe (ppm) | 350 |
| Cu (ppm) | 1.56 |
| Cd (ppm) | 0.00 |
| Cr (ppm) | 1.00 |
| Ni (ppm) | 1.4 |
| Pb (ppm) | 6.4 |

**Table 2.** Physical characteristics of soil used in this research.

| Properties | Soil |
|---|---|
| Skeletal fraction (%) | 68 |
| Sand (%) | 17 |
| Silt (%) | 11 |
| Clay (%) | 4 |
| Bulk density (g cm$^{-3}$) | 1.1 |
| Porosity (%) | 46.2 |
| Particle density (g cm$^{-3}$) | 2.3 |

*2.3. Growth Characteristics*

An pilot research project showed that the crate has sufficient space for root growth and distribution. Eight months after transplanting, twelve plants of each species were randomly chosen for root traits and root system observations. Root collar diameter (RCD) was measured. The intact root systems were cautiously removed from soil with flushing water. Root distribution and numbers were measured and recorded. Root area ratios (RAR) were calculated by gathering all roots (1–10 mm diameter) from every 10 cm soil layer utilizing Böhm's Methods [29] and calculated [17]. Root traits were inspected and root pictures were recorded for root architecture analysis. Root traits were measured using

WinRHIZO Pro image analysis software (v. 2009c; Regent Inc., Quebec, QC, Canada) [30], although root volume was assessed utilizing a water displacement method for accuracy [31]. Root and shoot were dried at 70 °C for 72 h for dry weight measurements. The acquired data were used to calculate root characteristics [32]. Meanwhile, live roots were collected for later tensile tests.

### 2.4. Uprooting Test

Twelve plants of each species were randomly selected for pullout capacity measurements. The silty clay soil has an average dry weight of 17.5 kN m$^{-3}$, and moisture content of $25 \pm 4\%$. Before each uprooting test, root collar diameter was recorded. The stem was cut off from 15 cm above the root collar. The uprooting test was carried out using a vertical pullout apparatus [17]. The ultimate uprooting force ($F_{ult}$, N) and displacement were recorded for further statistical analysis.

### 2.5. Root Tensile Test

After root system removal, single live root samples were cleaned and categorized into three diameter classes (0–1, 1–2, and 2–5 mm). Only intact root segments were used. Root segments were cut into 50 mm in length and preserved [33]. Seventy root sections of each species were tested in 24 h. The tests were carried out utilizing a tensile testing machine [34]. Prior to tensile test, root diameter at the middle of section was recorded. Root sections were tightened to clamps. Afterwards, the root sections were subjected to constant tensile speed of 4.7 mm min$^{-1}$ until rupture. For each species, 70 root sections were tested in the middle section: 39 root sections of Alpine silver grass and 36 root sections of Yushan cane plants. Root tensile strength ($T_{si}$, MPa) was calculated according to the equation [20,35]:

$$T_{si} = \frac{4F_{ult}}{\pi d_i^2} \tag{1}$$

where $F_{ult}$ is the ultimate tensile force at rupture (N), and $d_i$ is the root section diameter (mm) estimated at the middle point.

Furthermore, the relation between root tensile strength ($T_s$) and diameter ($d$) was computed according to simple power law equation [36]:

$$T_s = \alpha \cdot d^{-\beta} \tag{2}$$

where $\alpha$ and $\beta$ are species-specific coefficients.

### 2.6. Hydraulic Flume Experiment

Hydraulic flume experiments on simulated concentrated flow were conducted on eight-month-old plants using a flume similar to Burylo et al. [27]. The flume was constructed with stainless steel ($500 \times 30 \times 30$ cm, L × W × H) with an opening ($30 \times 30$ cm, L × W) at the bottom, equaling the wooden short crate, in order that the crate soil surface matches the flume surface (Figure 3). Before the flume test, the above-ground parts of samples were cut off at the base. Samples were immersed with water in a bucket for 2 h, and then left to drain for 10 h. For preventing side effects, the rims between crate and flume were sealed by adhesive tape and silicone glue. The surface slope, flow discharge and average bed flow discharge were monitored, and bed flow shear stress ($\tau$, Pa) was estimated according to the following formula [37,38].

$$\tau = \rho_w g R S \tag{3}$$

where $\rho_w$ is the water density (kg m$^{-3}$), g is the acceleration due to gravity (m s$^{-2}$), R is hydraulic radius (m) and S is the sin $\alpha$ in which $\alpha$ is soil surface slope (°). Subsequently, samples were subjected to a constant flow discharge of 0.55 m$^3$ m$^{-1}$ for one minute. Pilot tests showed that soil erosion mostly happened in the first minute of the flume test. Twelve plant sample crates of each species were tested at two slopes. Twelve crates with bare soil

samples were tested as control. Water and soil debris were gathered in 60 L metal buckets every minute at the outlet of the flume. The soil debris was divided by settling for 12 h, drying at 75 °C for 72 h and then measuring. The relative soil detachment rate of each species was computed by dividing the mass of soil debris removed from plant samples with the mean weight of soil debris detached from control ones. The tests consisted of two species and one control at two slopes, each with twelve replicates.

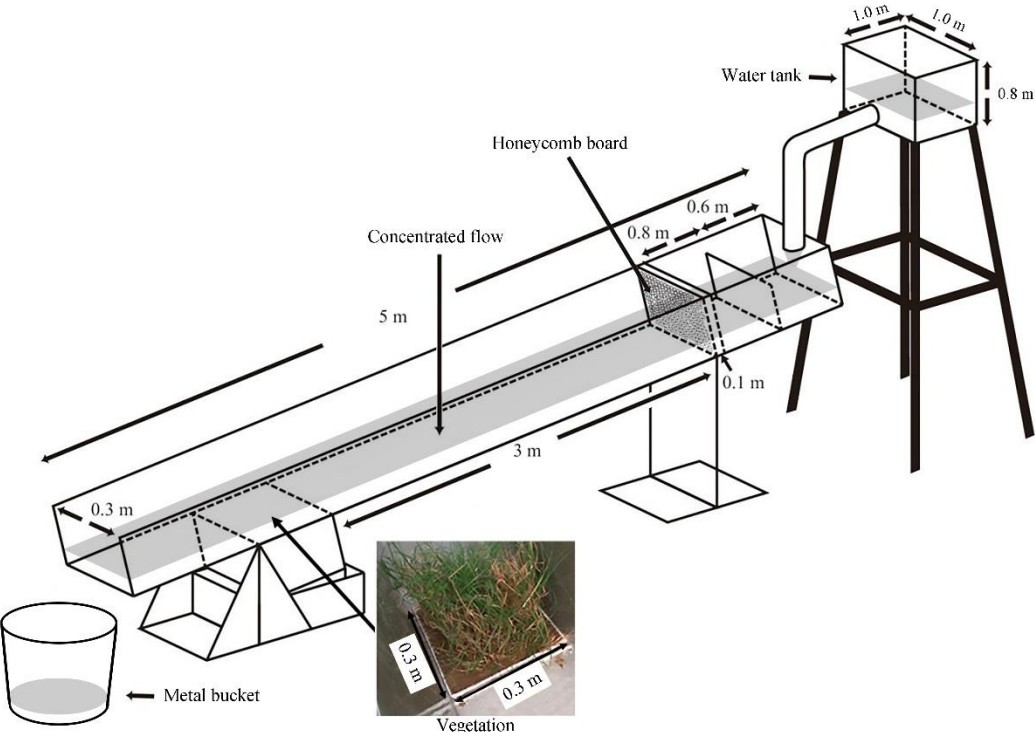

**Figure 3.** Schematic diagram of hydraulic flume test.

## 2.7. Statistical Analysis

Variations in root traits, root mechanical characteristics and water erosion-reducing ability between species were examined with *t*-test in IBM SPSS V22.0 (SPSS, Chicago, IL, USA). Data of root collar diameter and root surface area were normalized by $[(x_i - x^-)/SE]$, examined by multicollinearity tests, and multiple linear regression analysis utilizing SPSS Version 22.0. Multiple regression analyses in SPSS were conducted utilizing Multiple Regression Analysis in SPSS to inspect the relations between uprooting resistance and root traits. Microsoft Excel Regression analysis (Excel 2013, Microsoft, Redmond, WA, USA) was used to examine the relations between root tensile resistance, tensile strength, and root diameter.

## 3. Results

### 3.1. Root System Architecture

*Y. niitakayamensis* plants developed longer and more abundant root systems than *M. transmorrisonensis* plants (Figure 4). The fibrous root systems for *M. transmorrisonensis* and *Y. niitakayamensis* were classified as M-type [12]. In addition, *M. transmorrisonensis* had underground stems, and *Y. niitakayamensis* grew perennial sympodial rhizomes. *M. transmorrisonensis*, a grass, grew its fibrous roots and underground stems to 17 cm deep in soil (Figure 4a), while *Y. niitakayamensis*, a dwarf bamboo, developed its fibrous roots and rhizome up to 19 cm deep in soil (Figure 4b). Root area ratio (RAR) analysis revealed significant differences between species in top layers of soil (Table 3, Figure 5).

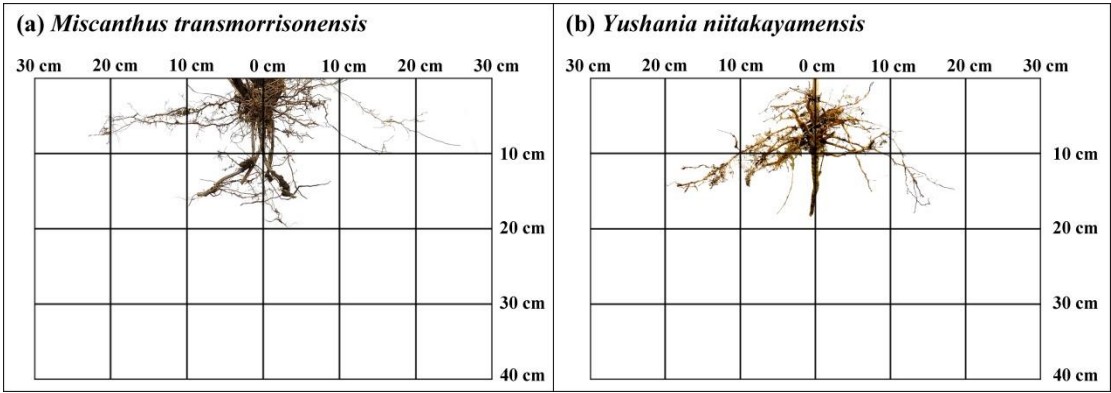

**Figure 4.** Root architecture of 8-month-old *M. transmorrisonensis* (**a**) and *Y. niitakayamensis* (**b**) plants.

**Table 3.** Means ± SDs of root area ratios at different soil depths for *M. transmorrisonensis* and *Y. niitakayamensis*.

| Species | Root Area Ratio (%) | | | | |
|---|---|---|---|---|---|
| | **0–5 cm** | **5–10 cm** | **10–15 cm** | **15–20 cm** | **20–25 cm** |
| *Miscanthus transmorrisonensis* | 0.17 ± 0.04 [a] | 0.08 ± 0.03 [b] | 0.02 ± 0.01 [b] | 0.01 ± 0.01 [a] | 0.00 ± 0.00 [a] |
| *Yushania niitakayamensis* | 0.26 ± 0.02 [a] | 1.13 ± 0.20 [a] | 0.17 ± 0.04 [a] | 0.04 ± 0.02 [a] | 0.00 ± 0.00 [a] |

Letters in the same column display significant difference (*t*-test) between species. N = 12. Level of significance $p < 0.05$.

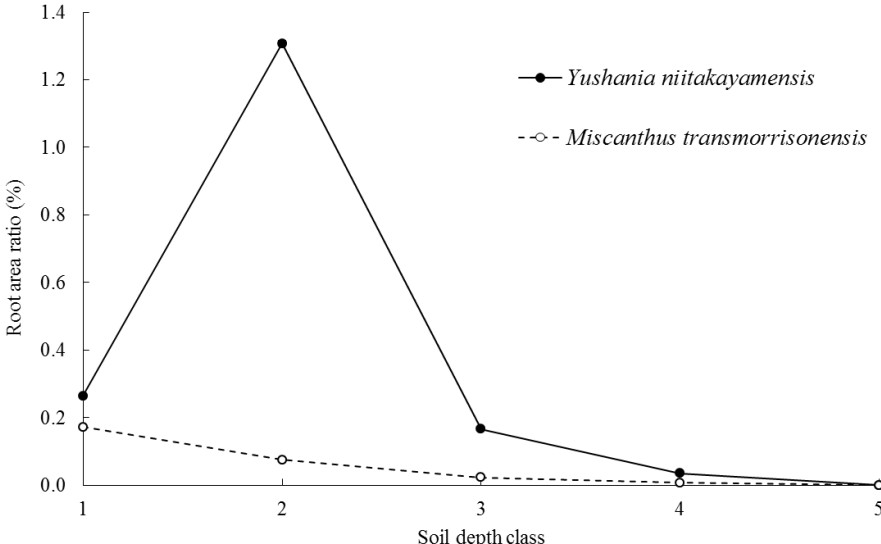

**Figure 5.** Root area ratios at different soil depths for *M. transmorrisonensis* and *Y. niitakayamensis*. Class: 1 (0–5 cm), 2 (5–10 cm), 3 (10–15 cm), 4 (15–20 cm), 5 (20–25 cm).

### 3.2. Growth Characteristics

Pilot testing revealed the wooden crates had enough room for root development during the experiment. Statistical results revealed great variations in root traits of plants between species (Table 4). Substantially, all growth characteristics except root collar diameter were significantly larger for *Y. niitakayamensis* than for *M. transmorrisonensis*. Altogether, *Y. niitakayamensis* plants exhibited significantly higher root and shoot growth characteristics than *M. transmorrisonensis* plants.

**Table 4.** Means ± SDs of growth characteristics for *M. transmorrisonensis* and *Y. niitakayamensis*.

| Growth Characteristics | *M. transmorrisonensis* | *Y. niitakayamensis* | t Value |
|---|---|---|---|
| RCD (mm) | 4.07 ± 0.21 [b] | 9.17 ± 0.58 [a] | 8.228 *** |
| RT | 1371.65 ± 130.28 [a] | 1125.78 ± 175.2 [a] | −1.118 |
| TRL (cm) | 332.08 ± 73.36 [b] | 1369.6 ± 121.14 [a] | −7.326 *** |
| RB (g) | 5.84 ± 1.12 [b] | 13.33 ± 2.65 [a] | 3.066 ** |
| SB (g) | 3.73 ± 0.39 [b] | 38.56 ± 3.59 [a] | 9.653 *** |
| RD (kg m$^{-3}$) | 2.33 ± 0.45 [b] | 5.64 ± 1.39 [a] | 2.263 * |
| RLD (km m$^{-3}$) | 1.2 ± 0.27 [b] | 5.48 ± 0.48 [a] | −7.705 *** |
| RSA (cm$^2$) | 333.68 ± 64.12 [b] | 1271.83 ± 161.03 [a] | −5.412 *** |
| RTD (g cm$^{-3}$) | 0.24 ± 0.02 [a] | 0.35 ± 0.06 [a] | 1.773 |
| RV (cm$^3$) | 26.47 ± 5.03 [b] | 45.23 ± 3.15 [a] | 1.855 * |
| SRL (m g$^{-1}$) | 0.33 ± 0.1 [b] | 3.48 ± 0.51 [a] | −6.052 *** |

RCD, root collar diameter; RT, root tips; TRL, total root length; RB, root biomass; SB, shoot biomass; RD, root density; RLD, root length density; RSA, total root surface area; RTD, root tissue density; RV, root volume; SRL, specific root length (SRL). Letters in the same row display significant differences (*t*-test) between species. N = 12. Level of significance * $p < 0.05$, ** $p < 0.01$, *** $p < 0.001$.

### 3.3. Root Anchorage Ability

Uprooting statistics exhibited that uprooting resistance force rose with uprooting up to a top and then fell as roots ruptured. The maximum uprooting resistance force of *Y. niitakayamensis* (0.8 ± 0.09 kN) was almost seven times that of *M. transmorrisonensis* (0.11 ± 0.02 kN) (Table 5). Regression results demonstrated a positive relation between the maximum uprooting force and some root traits, such as root collar diameter and root surface area. Linear regressions of uprooting force (U$_r$) and root collar diameter (RCD) for *M. transmorrisonensis* and *Y. niitakayamensis* are shown in Table 6. These results showed that uprooting resistance forces are strongly associated with root collar diameter and root surface area.

**Table 5.** Means ± SDs of ultimate uprooting force for *M. transmorrisonensis* and *Y. niitakayamensis* and t-value for a *t*-test.

| Root Anchorage Ability | *M. transmorrisonensis* | *Y. niitakayamensis* | t-Value |
|---|---|---|---|
| Ultimate uprooting force (kN) | 0.11 ± 0.02 [b] | 0.8 ± 0.09 [a] | 7.607 *** |

Letters signify significant difference (*t*-test) between species. N = 12. Significance level *** $p < 0.001$.

**Table 6.** Relationship between root traits and uprooting resistance for *M. transmorrisonensis* and *Y. niitakayamensis*.

| Morphological Traits | Species | Regression Equation | R$^2$ | p |
|---|---|---|---|---|
| RCD (mm) | *M. transmorrisonensis* | U$_r$ = 0.05RCD − 0.093 | 0.762 ** | 0.01 |
| | *Y. niitakayamensis* | U$_r$ = 0.141RCD − 0.509 | 0.8218 ** | 0.01 |
| RSA (cm$^2$) | *M. transmorrisonensis* | U$_r$ = −0.0004RSA + 0.05 | 0.738 * | 0.023 |
| | *Y. niitakayamensis* | U$_r$ = −0.001RSA + 0.407 | 0.803 * | 0.034 |

U$_r$, uprooting resistance; RCD, root collar diameter; RSA, root surface area. N = 12. Significance level * $p < 0.05$; ** $p < 0.01$.

Multicollinearity diagnostic test exhibited that variance inflation factors (VIF) of root collar diameter and root surface area for *M. transmorrisonensis* and *Y. niitakayamensis* were 1.139 and 1.078, respectively, indicating no collinearity between root collar diameter and root surface area. The derived multiple linear regression equations are shown in Table 7. Collectively, the root anchorage ability of *Y. niitakayamensis* is remarkably higher than that of *M. transmorrisonensis*.

**Table 7.** Relationship between uprooting resistance, root collar diameter and root surface area for *M. transmorrisonensis* and *Y. niitakayamensis*.

| Species | Regression Equation | $R^2$ | *p* | VIF |
|---|---|---|---|---|
| *M. transmorrisonensis* | Ur = 0.042RCD + 0.00028RSA − 0.097 | 0.715 ** | 0.002 | 1.139 |
| *Y.niitakayamensis* | Ur = 0.122RCD + 0.00RSA − 0.581 | 0.85 *** | 0.0004 | 1.078 |

U$_r$, uprooting resistance; RCD, root collar diameter; RSA, root surface area. N = 12. Significance level ** *p* < 0.01; *** *p* < 0.001.

### 3.4. Root Tensile Strength

The results revealed significant differences in root diameter, tensile resistance and tensile strength between the two species. The mean root diameter of *Y. niitakayamensis* was remarkably higher than that of *M. transmorrisonensis* and the mean root tensile resistance force of *Y. niitakayamensis* was also notably greater than that of *M. transmorrisonensis*. However, the average root tensile strength of *M. transmorrisonensis* was remarkably higher than that of *Y. niitakayamensis* (Table 8). In addition, root tensile resistance rose with increasing root diameter following power law function (Figure 6). However, root tensile strength declined with rising root diameter conforming to power law function (Figure 7). Taken together, the root tensile strength of *M. transmorrisonensis* was remarkably greater than that of *Y. niitakayamensis*.

**Table 8.** Means ± SDs of root diameter, root tensile resistance force and root tensile strength for *M. transmorrisonensis* and *Y. niitakayamensis* and t-value for a *t*-test.

| Parameters | *M. transmorrisonensis* | *Y. niitakayamensis* | t Value |
|---|---|---|---|
| Root diameters (mm) | 0.71 ± 0.06 [b] | 1.73 ± 0.17 [a] | −5.608 *** |
| Tensile resistance force (N) | 15.30 ± 1.94 [b] | 61.25 ± 8.96 [a] | −5.012 *** |
| Tensile strength (MPa) | 408.55 ± 22.71 [a] | 24.96 ± 1.13 [b] | 16.871 *** |

Letters in the same row display significant differences (t-test) between species. N = 12. Level of significance *** *p* < 0.001.

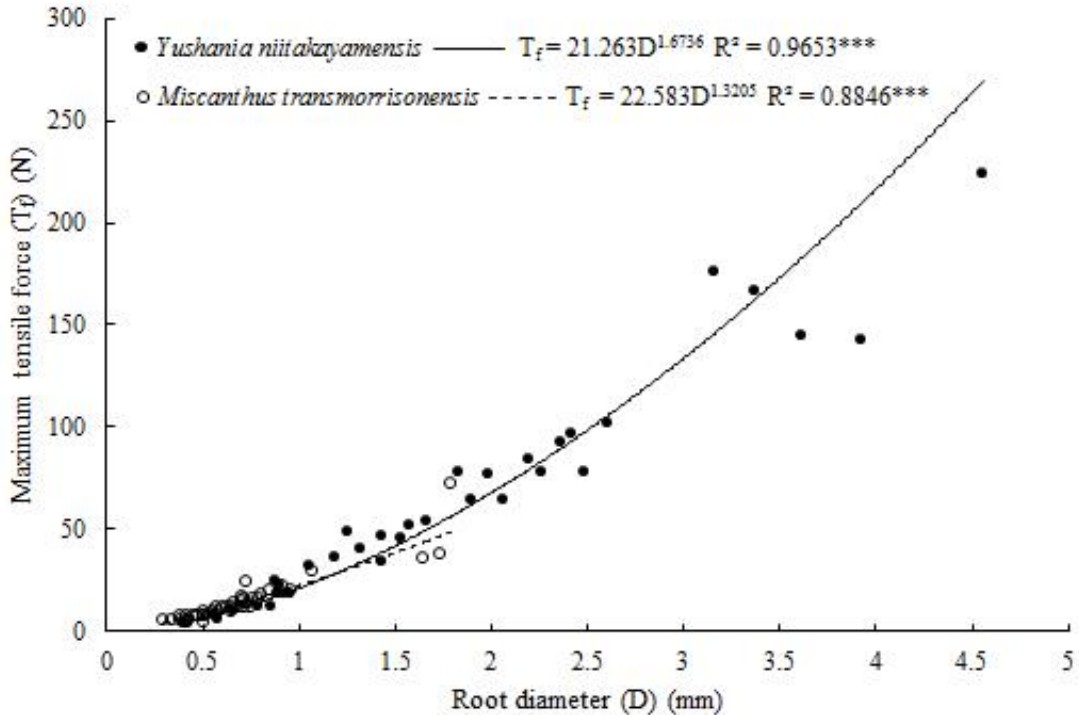

**Figure 6.** Root tensile resistance–root diameter relationship for the two alpine species. Level of significance *** *p* < 0.001.

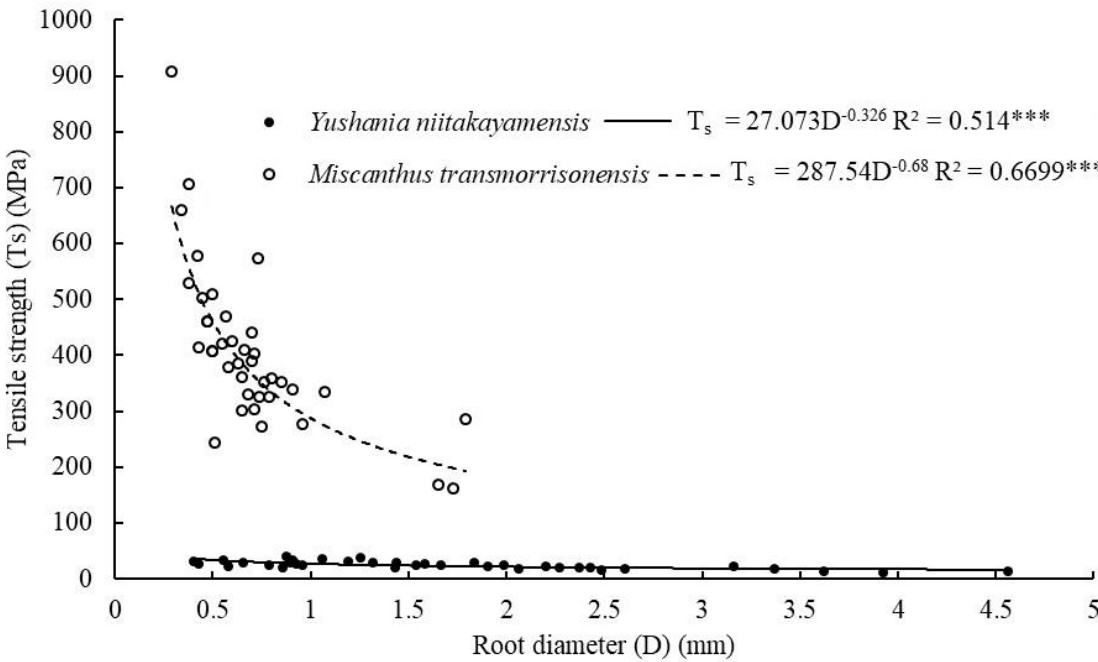

**Figure 7.** Root tensile strength–root diameter relationship for the two alpine species. Level of significance *** $p < 0.001$.

### 3.5. Water Erosion-Reducing Ability

Two soil surface slopes (2.86°, 16.7°) were used to test the water erosion-reducing ability by the two species in this study. For the slope of 2.86°, the mean flow velocity, flow discharge, and bottom flow shear stress were 47.49 cm s$^{-1}$, 0.55 l s$^{-1}$ and 0.0024 Pa, respectively. In addition, for the slope of 16.7°, the mean flow velocity, flow discharge, and bottom flow shear stress were 91.43 cm s$^{-1}$, 0.55 l s$^{-1}$ and 0.0083 Pa, respectively. Our results demonstrated that the soil loss amount was significantly varied among root-infiltrated soil samples of the two species and bare soil sample. At the slope of 2.86°, the mean soil loss amount for bare soil was significantly higher than that of root-permeated soil samples of *M. transmorrisonensis* and *Y. niitakayamensis*. Furthermore, at the slope of 16.7°, the mean soil loss amount for bare soil was significantly higher than that of root-permeated soil samples of *M. transmorrisonensis* and *Y. niitakayamensis* (Table 9). Moreover, relative soil detachment rates were significantly different between the two species. At the slope of 2.86°, the mean relative soil detachment rate of *M. transmorrisonensis* was at least six times higher than that of *Y. niitakayamensis*. At the slope of 16.7°, the mean relative soil detachment rate of *M. transmorrisonensis* was at least three times higher than that of *Y. niitakayamensis* (Table 10). Furthermore, the relative soil detachment rates (RSD) of the two species under two slopes decreased with rising root density (RD) conforming to power law function (Figures 8 and 9). Collectively, our results clearly show that *Y. niitakayamensis* has a superior water erosion-reducing ability compared to *M. transmorrisonensis*, as demonstrated in their reducing soil detachment rates.

**Table 9.** Means ± SDs of soil loss amount for *M. transmorrisonensis* and *Y. niitakayamensis* and bare soil.

| Slope (°) | Soil Loss Amount (g min⁻¹) | | | ANOVA (*p*) |
|---|---|---|---|---|
| | *Miscanthus transmorrisonensis* | *Yushania niitakayamensis* | *Bare Soil* | |
| 2.86 | 49 ± 10.02 [b] | 7.73 ± 1.15 [c] | 224.5 ± 41.5 [a] | 0.000 *** |
| 16.7 | 267.67 ± 35.45 [b] | 82.25 ± 12.61 [c] | 666.1 ± 66.3 [a] | 0.000 *** |

Letters in the same row display significant differences (ANOVA and Tukey's HSD post hoc test) among species. N = 12. Significance level *** $p < 0.001$.

**Table 10.** Means ± SDs of relative soil detachment rates between *M. transmorrisonensis* and *Y. niitakayamensis*.

| Slope (°) | Relative Soil Detachment Rate (%) | | *p* |
|---|---|---|---|
| | *Miscanthus transmorrisonensis* | *Yushania niitakayamensis* | |
| 2.86 | 21.83 ± 4.46 [a] | 3.44 ± 0.51 [b] | 0.005 ** |
| 16.7 | 40.19 ± 5.32 [a] | 12.35 ± 1.89 [b] | 0.003 ** |

Letters in the same row signify significant difference (T test) between species. N = 12. Significance level ** $p < 0.01$.

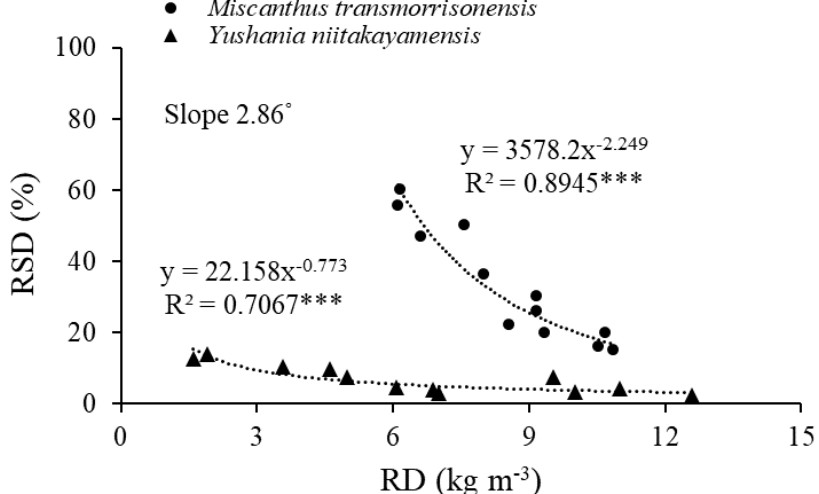

**Figure 8.** Relations between root density (RD) and relative soil detachment rate (RSD) at slope 2.86°. Level of significance *** $p < 0.001$.

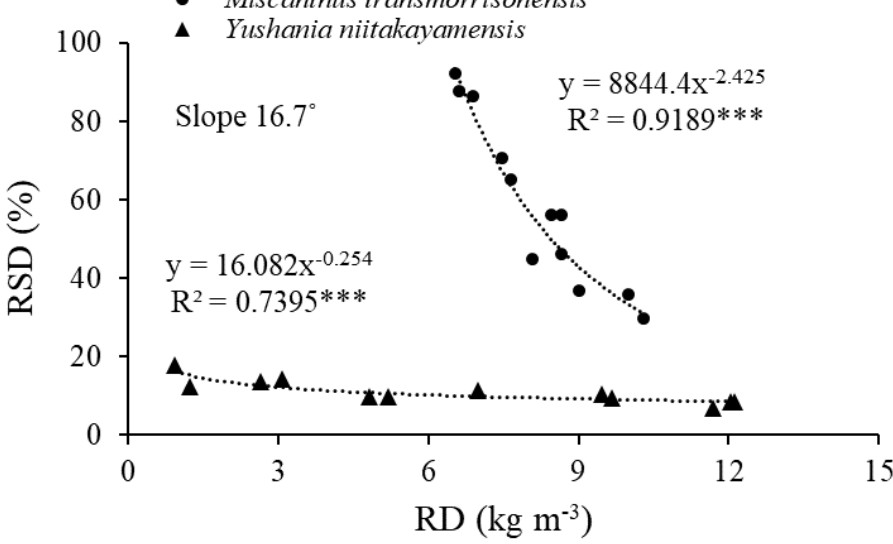

**Figure 9.** Relations between root density (RD) and relative soil detachment rate (RSD) at slope 16.7°. Level of significance *** $p < 0.001$.

## 4. Discussion

### 4.1. Root System Architecture

*M. transmorrisonensis* and *Y. niitakayamensis* are dominant native pioneer species occurring in post-fire alpine grassland in Taiwan. *M. transmorrisonensis* is a perennial grass exhibiting allelopathic dominance [8,39]. On the other hand, *Y. niitakayamensis* is a perennial sympodial bamboo species with a well-developed rhizome, which is more resistant to wildfire [40]. Our results showed that the roots of *M. transmorrisonensis* resemble an M- (massive) type fibrous root system with underground stems, while *Y. niitakayamensis* (dwarf bamboo) plants present an M- (massive) type fibrous root system with sympodial rhizomes. Previous studies indicate that plants with fibrous root systems are beneficial for erosion and sediment control [15,41]. Therefore, *M. transmorrisonensis* and *Y. niitakayamensis* with fibrous root systems are advantageous for soil conservation of alpine grassland. Further, the RAR distribution showed that *Y. niitakayamensis* has more roots in soil depths 5–10 cm and 10–15 cm than that of *M. transmorrisonensis*. Thus, our results suggest that *Y. niitakayamensis* is more competitive than *M. transmorrisonensis* in shallow soil erosion control.

### 4.2. Growth Characteristics

The results revealed that all root traits, except for root tips and root tissue density, varied notably between the two species. They were remarkably higher for *Y. niitakayamensis* than for *M. transmorrisonensis*. Previous studies have shown that root biomass, root density, root length density and total root surface area have a great influence on slope stability, soil erodibility and soil conservation [16,27,42,43]. On the post-fire alpine lands in Taiwan, vegetation restoration is crucial for soil conservation and ecological rehabilitation. *M. transmorrisonensis* and *Y. niitakayamensis* are native pioneer alpine species with adaptability to withstand harsh environments in alpine areas and are favorable for ecological restoration of alpine lands. Altogether, our results demonstrate that *Y. niitakayamensis* has better root growth characteristics and can adapt better to the harsh environments in the alpine areas than *M. transmorrisonensis*.

### 4.3. Root Anchorage Ability

The results demonstrated that the ultimate pullout force of bamboo *Y. niitakayamensis* is remarkably higher than that of grass *M. transmorrisonensis*. Regression analysis demonstrates strong correlations between uprooting resistance, root collar diameter and root surface area. These results are consistent with earlier studies [16,17,44]. Obviously, *Y. niitakayamensis* with sympodial rhizome and profuse fibrous roots has higher anchorage ability than *M. transmorrisonensis* with fibrous roots. Taken together, *Y. niitakayamensis* has the higher anchorage ability than *M. transmorrisonensis* and is more beneficial for soil conservation of alpine grasslands.

### 4.4. Root Tensile Strength

Root tensile strength has a great influence on slope stabilization [20,45,46]. Our findings demonstrated that tensile resistance force and tensile strength differ remarkably between the two species. Root tensile resistance force of *Y. niitakayamensis* was notably greater than that of *M. transmorrisonensis*, whereas root tensile strength of *M. transmorrisonensis* was remarkably greater than that of *Y. niitakayamensis*. Further, root tensile resistance is positively correlated with root diameter, congruent with previous studies [20,47,48]. However, the results show a negative power law correlation between root diameter and root tensile strength, consistent with previous studies [48,49]. These relations have been ascribed to decreasing cellulose content concurrent with rising root diameter [48] and decreasing lignin content as the root diameter rises [49]. Additional studies are needed to justify the root cellulose and lignin contents of the two alpine species.

*4.5. Water Erosion-Reducing Ability*

In general, alpine plants can reduce concentrated flow erosion [27]. *M. transmorrisonensis* and *Y. niitakayamensis* are native alpine pioneer plants and can withstand the alpine harsh environments. They can prevent water erosion by reducing soil detachment rates. Our study highlights the difference between the two alpine plant species on the reduction in soil detachment rates with respect to different slopes. The mean soil loss amount for bare soil is the highest, that of root-permeated soil samples of *M. transmorrisonensis* is the second, and *Y. niitakayamensis* is the lowest with respect to both slopes 2.86° and 16.7° (Table 6). In addition, the soil detachment rate of *M. transmorrisonensis* is remarkably higher than that of *Y. niitakayamensis*. Our results also show that the root density of *Y. niitakayamensis* is notably higher than that of *M. transmorrisonensis* and the relative soil detachment rates decrease with rising root density. Previous studies have demonstrated that a rise in root density results in a significant reduction in water erosion [22,26]. Altogether, our results clearly show that the water erosion-reducing ability of *Y. niitakayamensis* is significantly superior to that of *M. transmorrisonensis*.

In Taiwan, wildfires and torrential rains frequently trigger serious soil erosion and impact on alpine grassland. Vegetation restoration has become an important issue of sustainable alpine grassland management [50]. *M. transmorrisonensis* and *Y. niitakayamensis* are dominant native alpine pioneer species on post-fire grassland. These two species play an important role in vegetation restoration and ecological succession. Our findings indicate that there are remarkable differences in root characteristics, root anchorage ability, tensile strength and water erosion-reducing ability between these two native species, indicating that *Y. niitakayamensis* is superior to *M. transmorrisonensis* in restoring vegetation in post-fire alpine grasslands. Further studies are required to investigate the ecological succession of the wildfire burn scars. Silvicultural planting with native tree species can be adopted to foster forest ecosystem resilience and stability.

## 5. Conclusions

This study reveals that *Y. niitakayamensis* has remarkably better root growth characteristics and root anchorage ability than *M. transmorrisonensis*. Above all, hydraulic flume tests demonstrate that *Y. niitakayamensis* has superior water erosion-reducing ability than *M. transmorrisonensis*. These findings are beneficial for improving bioengineering technology of alpine grassland by combining the information of plant water erosion-reducing ability. It is clear that *Y. niitakayamensis* is beneficial for restoring vegetation of post-fire alpine grasslands. We also recommend that mixed planting with native tree species, such as *Abies kawakamii* and *Tsuga chinensis*, maybe practiced to foster forest ecosystem resilience and sustainability.

**Author Contributions:** J.-T.L. created and planned this study; S.-M.T., Y.-J.W., Y.-S.L. and M.-Y.C. investigated and collected data. S.-M.T. and Y.-J.W. and Y.-S.L. performed the data analysis; S.-M.T. and Y.-J.W. wrote the first draft; J.-T.L. wrote the final draft; J.-T.L. and M.-J.L. edited and revised the manuscript. All authors have read and agreed to the published version of the manuscript.

**Funding:** This research was financed by the Ministry of Science and Technology of Taiwan Grant No. MOST 109-2311-B-415-001.

**Institutional Review Board Statement:** Not applicable.

**Informed Consent Statement:** Not applicable.

**Acknowledgments:** We thank Maurice S. B. Ku from Department of Bio-agricultural Science, National Chiayi University for English editing, valuable suggestions and reviewing this paper.

**Conflicts of Interest:** The authors declare no conflict of interest.

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
