# Peer review of "Root Characteristics and Water Erosion-Reducing Ability of Alpine Silver Grass and Yushan Cane for Alpine Grassland Soil Conservation"

_sustainability, doi:10.3390/su13147633_

Round 1
Reviewer 1 Report
The manuscript, titled "Root Characteristics and Water Erosion Reducing
Ability of Alpine Silver Grass and Yushan Cane for Alpine Grassland Soil Conservation" is interesting in some aspects of soil protection of erosions with use some native plants, which able to stabilize soil surface. Nevertheless, I would not like to reccomend this paper to publication in Sustainability journa, because in current form this paper does not reflect regional or global context of sustainability. Data, presented in paper are raw and novel, but not logically dealed with sustanability of bioshpere or ecosystems. This was general comments. I also have numerous minors, here they are:
-lack of data on soil - "The soil pedon has two main lithological discontinuities at about 40 and 70 cm 82 below soil surface. The soil materials of upper 45 cm and 45-70 cm are silty clay and sandy loam, respectively" - So , if we have soil vertical unhomogenity it should be more detailed interpreted in terms of slope in soil erosion, also the type of soil according WRB should be provided, soil picture should be given as well.
-if we are speaking about root ability to penetrate to soil and to prevent erosions we have to obtain data on soil physics - soil porosity, soil density, full partcile size distribution etc. Table 1 is good, but it is completely chemical. I think that table 2 with Key soil physical properties will be usefull.
-soil map of the studied plots should be provided
-citation of literature on postfire soil is regional, without taking into account results many previous studies of soils under the fire effect in different regions of the Earth
In current version, paper more fit to some agorphysical journal.
Author Response
Response to Reviewer 1
sustainability-1249318
Reviewer 1: Comments and Suggestions
The manuscript, titled "Root Characteristics and Water Erosion Reducing
Ability of Alpine Silver Grass and Yushan Cane for Alpine Grassland Soil Conservation" is interesting in some aspects of soil protection of erosions with use some native plants, which able to stabilize soil surface. Nevertheless, I would not like to reccomend this paper to publication in Sustainability journa, because in current form this paper does not reflect regional or global context of sustainability. Data, presented in paper are raw and novel, but not logically dealed with sustanability of bioshpere or ecosystems. This was general comments.
Response 1: We greatly appreciate the reviewer’s constructive comments and valuable suggestions. It is true that this paper does not reflect global context, albeit it is a typical case of post-fire rehabilitation, soil restoration and vegetation recovery in the region of Southeast Asia. The data presented in this paper deal with sustainability of alpine ecosystems.
Point 2: I also have numerous minors, here they are:
-lack of data on soil - "The soil pedon has two main lithological discontinuities at about 40 and 70 cm 82 below soil surface. The soil materials of upper 45 cm and 45-70 cm are silty clay and sandy loam, respectively" - So , if we have soil vertical unhomogenity it should be more detailed interpreted in terms of slope in soil erosion, also the type of soil according WRB should be provided, soil picture should be given as well
Response 2: Thanks to the reviewer for the constructive comments and valuable suggestions. The genesis and classification of soils developed under Yusan cane grassland in Hohuan Mountain areas (the study site) have been reported by Dr. H.B. King. This has been shown in Reference 28 “King, H.B. Genesis and classification of soils developed under Yushan cane (Yushania niitakayamensis) grassland in the Hohuan Mountain area. Bull. Taiwan For. Res. Inst. New Series 1993, 8, 21–38”. The slope in soil erosion has been described in 2.1. Site and vegetation (L82-83). The type of soil was classified as “typic Haplumbrept, fine, illitc, frigid” according to Dr. King’s report. This has been revised in the manuscript accordingly. (L87-88)
Point 3: -if we are speaking about root ability to penetrate to soil and to prevent erosions we have to obtain data on soil physics - soil porosity, soil density, full partcile size distribution etc. Table 1 is good, but it is completely chemical. I think that table 2 with Key soil physical properties will be usefull.
Response 3: Thanks to the reviewer for the constructive comments and suggestions. Table 2 with key soil physical properties has been added to the materials and methods section and revised in the manuscript. (L116-117)
Point 4: -soil map of the studied plots should be provided
Response 4: Thanks to the reviewer for the suggestion. The soil profile of Yusan cane grassland has been documented in Dr. H.B. King’s paper “Reference 28 King, H.B. Genesis and classification of soils developed under Yushan cane (Yushania niitakayamensis) grassland in the Hohuan Mountain area. Bull. Taiwan For. Res. Inst. New Series 1993, 8, 21–38”. The soil profile of Yusan cane grassland showed an Umbric epipedon (A, at 0-22 cm) and two Cambic horizons (Bw1, at 22-32 cm and Bw2, at 32-44 cm). These has been added and revised in the manuscript accordingly. (L85-86)
Point 5: -citation of literature on postfire soil is regional, without taking into account results many previous studies of soils under the fire effect in different regions of the Earth
Response 5: Thanks to the reviewer for the constructive comments. It is true that the citation of literature on post-fire soil is regional. We focused on the wildfire effect on vegetation recovery, soil restoration and control of soil erosion in this region. Nowadays, control of soil erosion and degradation has become an important issue for post-fire management. Thus, this study aimed to investigate the root characteristics, biomechanical properties and water erosion reducing ability of these two native pioneer species for improving bioengineering technology of post-fire alpine grassland.

Reviewer 2 Report
Dear authors,
Your paper is a study that analyzes a lot of data, statistically assured, but in the future the analysis of several species could be taken into account, not only 2. Because in this paper we have only 2 values of the 2 species.
Introduction
L70-73 The aim of the study needs to be explained in a separate paragraph also, you need to state the hypothesis and objectives of the study in separate sentences in order to explain why are they important.
Materials and Methods
Add reference for climate and specify the dimension of your research area, maybe the percent of coverage of each species.
Results
Root System Architecture is a domain very interest and actuality but please do not repeat integral in the text the values from tables.
Make condensed explanations. Take 2 or more parameters with the same significance of differences and present them grouped.
3.3. Root Anchorage Ability
Describe the regressions. Do not repeat them entirely, make a table with them.
I suggest for the entire results section to make condensed tables to present more parameters in a smaller space. You need to say why you have the obtained results. In this form they look just like data sets. You have a lot of space to explain based on root traits the results.
Discussion
I recommend you do not make reference to tables or figure from results.
In conclusion, it is a work with many data and current results
Author Response
Response to Reviewer 2
sustainability-1249318
Reviewer 2: Comments and Suggestions
Your paper is a study that analyzes a lot of data, statistically assured, but in the future the analysis of several species could be taken into account, not only 2. Because in this paper we have only 2 values of the 2 species.
Response 1: We greatly appreciate the reviewer’s constructive comments and valuable suggestions. In the future, we will investigate more species appearing on the alpine grassland.
Point 2: L70-73 The aim of the study needs to be explained in a separate paragraph also, you need to state the hypothesis and objectives of the study in separate sentences in order to explain why are they important.
Response 2: Thanks to the reviewer for the valuable suggestions. This has been revised accordingly in the manuscript. (L71-75)
Point 3: Materials and Methods
Add reference for climate and specify the dimension of your research area, maybe the percent of coverage of each species.
Response 3: Thanks to the reviewer for the valuable suggestions. These have been added in the manuscript (L82-84).
Point 4: Results
Root System Architecture is a domain very interest and actuality but please do not repeat integral in the text the values from tables.
Make condensed explanations. Take 2 or more parameters with the same significance of differences and present them grouped.
Response 4: Thanks to the reviewer for the constructive comments and valuable suggestions. These have been revised in results section of the manuscript (L193-296).
Point 5: 3.3. Root Anchorage Ability
Describe the regressions. Do not repeat them entirely, make a table with them.
I suggest for the entire results section to make condensed tables to present more parameters in a smaller space. You need to say why you have the obtained results. In this form they look just like data sets. You have a lot of space to explain based on root traits the results.
Response 5: Thanks to the reviewer for the constructive comments and valuable suggestions. The regressions have been condensed into Tables 6 and 7. These have been revised in the manuscript (L280-287).
Point 6: Discussion
I recommend you do not make reference to tables or figure from results.
In conclusion, it is a work with many data and current results
Response 6: Thanks to the reviewer for the constructive comments and valuable suggestions. These have been revised in the manuscript (L367-422).

Reviewer 3 Report
In the manuscript, the authors compare root characteristics and water erosion reducing ability of two endemic species emerging on post-fire alpine grassland in Taiwan: Alpine silver grass (Miscanthus transmorrisonensis) and Yushan cane (Yushania niitakayamensis). Using detailed measurements, hydraulic flume experiments, and statistical analysis they prove that Yushan cane has superior root characteristics and water erosion reducing ability than Alpine silver grass and is thus more beneficial for restoring vegetation of post-fire alpine grasslands. The study addresses a topic that has not been extensively studied before providing new and valuable information that can be used for improving bioengineering technology of alpine grassland. In my opinion, the manuscript meets the requirements of the journal.
Author Response
Response to Reviewer 3
sustainability-1249318
Reviewer 3: Comments and Suggestions
In the manuscript, the authors compare root characteristics and water erosion reducing ability of two endemic species emerging on post-fire alpine grassland in Taiwan: Alpine silver grass (Miscanthus transmorrisonensis) and Yushan cane (Yushania niitakayamensis). Using detailed measurements, hydraulic flume experiments, and statistical analysis they prove that Yushan cane has superior root characteristics and water erosion reducing ability than Alpine silver grass and is thus more beneficial for restoring vegetation of post-fire alpine grasslands. The study addresses a topic that has not been extensively studied before providing new and valuable information that can be used for improving bioengineering technology of alpine grassland. In my opinion, the manuscript meets the requirements of the journal.
Response: We greatly appreciate the reviewer’s constructive comments and positive affirmation.

Round 2
Reviewer 1 Report
Authors have followed my comments and suggestions and paper become better sturctured and more logicallyy connected with topic of journal. Only minor suggestions have to be made before paper will be accepted for publication:
- Is is possible to add to soil name verificator "inclinic" whihc is typical for some slope soils? it is not obligatory, but authors have to consider this possibility
- table 2 "Gravel" , but no "Graval", but , here, in this ocntext it is better to call this fraction "Skeletal fraction"
- table 2 - Specific gravity - do you meanr density of solid soil phase?
- table 2 - summ of fraction is given for both - skeletal and fine earth, normally, in soil science we use to calculate silt, clay and sand in percents to fine earth. If one do not do it, he should explain why.
- Table 1 - you have to describe methods of nutrients determination, not only type of equipenemt, but, also , type of extragent.
- I suggest to provide insert map of investigated plot which will illustrate the location of plot in country.
Author Response
Response to Reviewer 1
(Round 2)
sustainability-1249318
Reviewer 1: Comments and Suggestions
Authors have followed my comments and suggestions and paper become better sturctured and more logicallyy connected with topic of journal. Only minor suggestions have to be made before paper will be accepted for publication:
Response: We sincerely appreciate the reviewer’s valuable comments and suggestions, which help us to improve the quality of our manuscript.
Point 1: Is is possible to add to soil name verificator "inclinic" whihc is typical for some slope soils? it is not obligatory, but authors have to consider this possibility
Response 1: We would like to thank the reviewer for the valuable suggestion. This has been revised in the manuscript accordingly. (L89)
Point 2: table 2 "Gravel" , but no "Graval", but , here, in this ocntext it is better to call this fraction "Skeletal fraction"
Response 2: Thanks to the reviewer for the constructive comments and suggestions. Yes, it is better to call this fraction “Skeletal fraction”. This has been revised in Table 2. (L126-127)
Point 3: table 2 - Specific gravity - do you meanr density of solid soil phase?
Response 3: Thanks to the reviewer for the valuable comment and suggestion. The specific gravity means density of solid soil phase. This has been revised in Table 2 as “Particle density”. (L126-127)
Point 4: table 2 - summ of fraction is given for both - skeletal and fine earth, normally, in soil science we use to calculate silt, clay and sand in percents to fine earth. If one do not do it, he should explain why.
Response 4: Thanks to the reviewer for the constructive comments. It is true that in soil science we use to calculate silt, clay and sand in percents to fine earth. However, this research is dealing with mountainous forest soils, which contain a lot of skeletal fraction (gravel). In this case, we calculate skeletal fraction, sand, silt and clay. We have revised the skeletal fraction in Table 2 accordingly.
Point 5: Table 1 - you have to describe methods of nutrients determination, not only type of equipenemt, but, also, type of extragent.
Response 5: Thanks to the reviewer for the constructive comments and valuable suggestions. These have been revised in the manuscript as “The organic carbon and total nitrogen contents were analyzed by combustion method using Vario EL cube elemental analyzer. In addition, Mehlich 3 extragent (composing 0.2 M glacial acetic acid, 0.25 M ammonium nitrate, 0.015 M ammonium fluoride, 0.013 M nitric acid, and 0.001 M ethylene diamine tetraacetic acid (EDTA)) was used to extract the elements, such as P, K, Ca, Mg, Zn, Mn, Fe, Cu, Cd, Cr, Ni, and Pb. The concentrations of these elements were analyzed using inductively coupled plasma optical emission spectroscopy (ICP-OES) method.”. (L110-117)
Point 6: I suggest to provide insert map of investigated plot which will illustrate the location of plot in country.
Response 6: We sincerely thank the reviewer for the constructive comments and suggestions. This has been revised in Figure 2 as “Diagram of experimental plot location (a) and the sample plot layout (b)”. (L91-93)
